# Validity, Reliability, and Usefulness of My Jump 2 App for Measuring Vertical Jump in Primary School Children

**DOI:** 10.3390/ijerph17103708

**Published:** 2020-05-25

**Authors:** Špela Bogataj, Maja Pajek, Vedran Hadžić, Slobodan Andrašić, Johnny Padulo, Nebojša Trajković

**Affiliations:** 1Department of Nephrology, University Medical Centre, 1000 Ljubljana, Slovenia; spela.bogataj@kclj.si; 2Faculty of Sport, University of Ljubljana, 1000 Ljubljana, Slovenia; maja.pajek@fsp.uni-lj.si (M.P.); vedran.hadzic@fsp.uni-lj.si (V.H.); 3Faculty of Economics Subotica, University of Novi Sad, 2100 Novi Sad, Serbia; slobodan.andrasic@ef.uns.ac.rs; 4Department of Biomedical Sciences for Health, Università degli Studi di Milano, 20133 Milan, Italy; johnny.padulo@unimi.it; 5Faculty of Sport and Physical Education, University of Novi Sad, 21000 Novi Sad, Serbia

**Keywords:** young adolescents, jumping ability, assessment, physical education, testing

## Abstract

There is a persistent need in sport science for developing a measuring tool that is affordable, portable, and easy to use. We aimed to examine the concurrent validity and test–retest reliability of the My Jump 2 app compared to a validated OptoJump instrument for measuring jump performance during the squat jump (SJ), countermovement jump (CMJ), and CMJ free arms (CMJAM) in primary school children. A total of 48 participants (11–14 years age), volunteered to participate in this research. The jumps were recorded with a validated OptoJump photoelectric cell system and a concurrent device (iPhone X through My Jump 2 app) at the same time. The participants repeated the testing procedure after two weeks to assess the reliability of the measurements (ICC). Systematic bias between sessions and tools was evaluated using the paired samples *t*-test and Bland and Altman analysis. High test–retest reliability (ICC > 0.89) was observed for all measures’ in-between conditions. Very large correlations in the total sample were observed between the My Jump 2 app and OptoJump for SJ (r = 0.97, *p* = 0.001), CMJ (r = 0.97, *p* = 0.001), and CMJAM (r = 0.99, *p* = 0.001). Bland and Altman’s plot depicting limits of agreement for the total sample between the OptoJump and My Jump 2 show that the majority of data points are within the 95% CIs. The results of this study suggest that My Jump 2 is a valid, reliable, and useful tool for measuring jump performance in primary school children.

## 1. Introduction

Physical fitness was recognized as a key health marker in children and adolescents [1], which can also predict later health status [2]. One branch of physical fitness is called musculoskeletal fitness, which stands for motor fitness, muscular strength, flexibility, and muscular endurance [3]. Among youth, different levels of musculoskeletal fitness have been conditioned by physical education (PE) in schools [4], which aims to incorporate activities for enhancing children’s physical fitness level [5].

These findings suggest the need for physical fitness testing in children and adolescents and that regular testing and physical activity should be considered as a public health priority [6,7]. Most accurately, we can measure physical fitness through laboratory methods. However, due to the expensive and limited equipment in the clinical laboratory only, these tests are not carried out on the whole population field testing.

One of the most common tests for assessing physical fitness in different populations is the vertical jump (VJ) [8,9,10]. VJ is a common method used to evaluate the lower limb power/muscle strength in children [11] as well as in the PE class [12]. The VJ is simple to administrate and has important outcome information about children’s neuromuscular capabilities and their development process [13]. Squat jump (SJ), countermovement jump (CMJ), and drop jump (DJ) represent the VJ’s battery test. The most commonly used validated devices for measuring VJ height among professionals are force platforms, contact platforms, infrared platforms, accelerometers, and high-speed cameras [14,15,16]. However, due to their price, bulkiness, and need of a software skill to analyze the data, they are rarely used among children and adolescents. Indeed, for children a non invasive evaluation as a VJ is highly appreciated, therefore, we need a portable and inexpensive tool that is valid and reliable.

Recently, Apple Inc. (USA) released My Jump 2 application, which claims to calculate jump height by recording high-speed video directly. The app is affordable, practical, and can be used in different fields [17,18]. The validity of this app or its previous version (My Jump) has been demonstrated on the elderly [19], young recreational athletes [18,20], sport science students [17,21], trained athletes [22], and even on cerebral palsy football players [23]. However, the validity and reliability of the My Jump 2 app for the primary school population have not been tested yet. Therefore, this study aimed to verify the validity and reliability of the My Jump 2 app with an iPhone X for the evaluation of SJ, CMJ, and CMJAM in primary school children compared to a previously validated OptoJump photoelectric cells system [15]. It was hypothesized that the My Jump 2 app would have good test–retest reliability and concurrent validity compared to the Optojump device.

## 2. Materials and Methods

### 2.1. Participants

A total of 48 participants aged 11–14 years, volunteered to participate in the research. The sample consisted of 22 girls and 26 boys from the primary school “Jovan Mikic” in Subotica, Serbia (Table 1). The testing was performed in 2020 on February 28th and March 13th. On the day of testing, they were healthy, without any heart or pulmonary disease and injury-free. Before the testing, they were not involved in any strength, jumping, or high-intensity training for 48 h. They were informed and familiarized with the testing procedures, and before the start, their parents signed a written informed consent form. The research adhered to the Declaration of Helsinki and was approved by the local ethics committee (ref. 12/1041).

### 2.2. Procedures

All participants were familiarized with SJ, CMJ, and CMJAM techniques before testing. Physical education teachers have educated children of the proper technique one week in advance by video and live demonstration and the explanation of the correct technique. Before testing, they carried out a standard 10 min warm-up consisted of jogging, skipping, vertical jumps, and calisthenics. Their body mass was measured to the nearest 0.1 kg with electronic scale TANITA BC 540 (TANITA Corp., Arlington Heights, IL, USA) and body height with a stadiometer (SECA Instruments Ltd., Hamburg, Germany) to the nearest 1 cm [24]. Then, each participant performed three SJs, three CMJs, and three CMJAMs with the instruction to jump as high as possible. Between the trials, there was a two-minute passive rest. The highest jump of each technique was taken into analysis. The jumps were recorded with a validated OptoJump [25] system (Optojump, Microgate, Bolzano, Italy) and with an iPhone X (v.13, Apple Inc., Cupertino, CA, USA) through the My Jump 2 app concurrently. The participants repeated the testing procedure after two weeks with the same conditions randomly to assess the reliability of the measurements.

### 2.3. Squat Jump Performance

Participants were instructed to start the jump [26] in the position of the 90° knee flexion [27] with the feet a shoulder-width apart and with their hands on their hips. They were asked to jump for maximum height and maintain their hands on their hips [28]. Countermovement was discouraged, and in case of any mistake, the jump was repeated.

### 2.4. Countermovement Jump Performance

The CMJ starting position was a standing position with a straight torso and knees fully extended with the feet shoulder-width apart [29]. Participants were asked to keep their hands on their hips throughout the whole jump. They were instructed to perform a quick downward movement (approximately 90° of knee flexion), and afterward a fast-upward movement to jump as high as possible.

### 2.5. Countermovement Jump Free Arms Performance

The CMJAM technique is similar to CMJ with the exception of arm movement [30]. Participants are instructed to swing back with their arms during the downward movement and forward during the upward movement.

### 2.6. Optojump Photoelectric Cell System

The Optojump device is an infrared platform with proven validity and reliability for assessing vertical jumps heights. It achieved strong concurrent validity in comparison with the force platform (ICC = 0.99; 95% CI = 0.97–0.99; *p* < 0.001) and was recognized as a legitimate instrument for field-based vertical jump assessments [15].

### 2.7. My Jump 2 App

My Jump 2 for iPhone X was used to calculate the jump height by manually selecting the take-off frame and landing frame (Figure 1) of the video. The app determines the jump height using the equation *h* = *t*^2^ × 1.22625 described by Bosco et al. [31] where *h* stands for the jump height (in meters) and *t* for flight time (in seconds). All collections were made with the same phone and by the same evaluator with no professional experience in video analysis. The evaluator was always recording from the same position and with the same distance from the participants (1.5 m) as standard calibration according to the manufactory instructions.

### 2.8. Statistical Analysis

Descriptive statistics were presented using means and standard deviations. A Shapiro–Wilk test was used to check the data normality. Systematic bias between sessions and tools was evaluated using the paired samples *t*-test [15]. Standardized differences in mean (with 95% confidence intervals; CI) were calculated to determine the magnitude of the change across and between tests. According to Hopkins et al. [32], Cohen d effect size (ES) magnitudes of change were classified as trivial (>0.2), small (0.2–0.5), moderate (0.5–0.8), large (0.8–1.60), and very large (>1.60). Reliability between the test–retest was analyzed using intraclass correlation coefficient (ICC), typical error (TE) expressed as the coefficient of variation (CV%), and smallest worthwhile change (SWC) according to the Excel spreadsheet provided by Hopkins (2007) [33]. Regarding the ICC analysis, single measure, two-way mixed, absolute-agreement parameters were used [34]. The highest jump from each subject on both testing sessions, retrieved from My Jump 2, was used. ICC was interpreted as <0.1 = low, <0.3 = moderate, <0.5 = high, <0.7 = very high, <0.9 = nearly perfect, and <1.0 = perfect. Good reliability was considered if the following criteria was fulfilled: CV < 5% and ICC > 0.69 [35]. Test usefulness was determined based on the comparison of SWC (0.2 multiplied by the between-subject SD, based on Cohen’s ES) to TE [36]. The following criteria were used to establish the usefulness of tests: “Marginal” (TE > SWC), “OK” (TE = SWC), and “Good” (TE < SWC).

The concurrent My Jump 2 App validity was tested with Pearson’s product-moment correlation coefficient (r) and Bland and Altman analysis in which the difference between both devices was plotted against the mean of the two devices [37]. The statistical significance was fixed at the *p* ≤ 0.05 level.

## 3. Results

### 3.1. Reliability

Similar SJ (test = 22.3 ± 4.1 cm; retest = 22.8 ± 4.3 cm), CMJ (test = 24.5 ± 4.7; retest = 25.0 ± 5.1 cm), and CMJAM (test = 27.0 ± 5.8 cm; retest = 27.7 ± 6.0 cm) values were observed between testing sessions in primary school children. Non-significant differences (*p* > 0.05) were observed between testing sessions for SJ (ES = trivial; CI 95% (−0.1; 1.1)), CMJ (ES = trivial; CI 95% (0.0; 0.8)), and CMJAM (ES = trivial; CI 95% (−0.2; 0.7)) as observed in Table 2. High test–retest reliability (ICC > 0.88) was observed for all measures. However, TE (expressed as CV%) < 5% for the whole sample was observed for CMJ and CMJAM, respectively.

Table 2 shows the test–retest results for SJ (test = 23.2 ± 4.6 cm; retest = 23.5 ± 4.6 cm), CMJ (test = 25.1 ± 5.5 cm; retest = 25.7 ± 6.1 cm), and CMJAM (test = 28.2 ± 6.9 cm; retest = 29.2 ± 7.3 cm) in boys. There were no significant differences (*p* > 0.05) between testing sessions for SJ (ES = trivial; CI 95% (−0.8; 1.3)), CMJ (ES = trivial; CI 95% (−0.2; 1.2)), and CMJAM (ES = trivial; CI 95% (−0.3; 1.0)). High test–retest reliability (ICC > 0.89) was observed for all measures.

There was higher TE (expressed as CV%) only for SJ (8.3%), while for CMJ and CMJAM, TE was lower than 5%. Similar results for test–retest were observed for girls (ICC > 0.89), with TE (expressed as CV%) < 5% for all tests, respectively.

### 3.2. Test Usefulness

The TE for SJ for the whole sample, and for boys and girls separately, was greater than the presumed SWC; consequently, these measures were rated as “marginal”. In contrast, TE for CMJ and CMJAM for both genders were similar or lower than SWC and were rated as “OK” and “good” (see Table 2).

### 3.3. Concurrent Validity of the My Jump 2 App

There were no significant differences (*p* > 0.05) between the My Jump 2 app and OptoJump for all jumps in primary school children with trivial effects size (from −0.02 to 0.02; Table 3). Very large correlations in whole sample were observed between the My Jump 2 app and OptoJump for SJ (r = 0.97, *p* = 0.001), CMJ (r = 0.97, *p* = 0.001), and CMJAM (r = 0.99, *p* = 0.001). Similar results were obtained for boys and girls separately. No significant differences (*p* > 0.05) were observed between the My Jump 2 app and OptoJump for all jumps in boys and girls with trivial effects size (from −0.08 to 0.02). Very large correlations were observed between the My Jump 2 app and OptoJump for all jumps in boys (r > 0.98, *p* = 0.001) and girls (r > 0.97, *p* = 0.001), respectively.

Figure 2, Figure 3 and Figure 4 show limits of agreement between My Jump 2 and OptoJump measures of SJ, CMJ, and CMJAM. The charts indicate that 3/48 (6.3%), 1/48 (2.1%), and 3/48 (6.3%) of the data points were beyond the mean ± 1.96 SD lines for SJ, CMJ, and CMJAM, respectively. Bland and Altman’s plot depicting limits of agreement for SJ between the OptoJump and My Jump 2 show that the majority of data points were within the 95% CIs (Figure 2). The mean bias between the two methods for SJ was −0.48 cm.

The Bland–Altman plot for CMJ for the whole sample is presented in Figure 3. Bland and Altman’s plot depicting limits of agreement for CMJ between the OptoJump and My Jump 2 show that the majority of data points were within the 95% CIs (Figure 3). The mean bias between the two methods for CMJ was 0.13 cm.

The Bland–Altman plots regarding CMJAM height for the whole sample assessed with the two apparatus are illustrated in Figure 4. Bland and Altman’s plot depicting limits of agreement for CMJAM between the OptoJump and My Jump 2 show that the majority of data points were within the 95% CIs (Figure 4). The mean bias between the two methods for CMJAM was −0.05 cm. Further analysis of the Bland–Altman plots in primary school children revealed very low R^2^ values (R^2^ ≤ 0.12), meaning outcomes estimated from My Jump 2 had no predisposition to overestimate or underestimate jump performance.

## 4. Discussion

The present study examined the concurrent validity and test–retest reliability of My Jump 2 installed on an iPhone X compared to a validated OptoJump instrument, for measuring jump performance during SJ, CMJ, and CMJAM, in primary school children. My Jump 2 was found to be highly valid and reliable in measuring the jump height of an SJ, CMJ, and CMJAM in comparison with an OptoJump. Moreover, CMJ and CMJAM tests showed it to be practically useful to assess and monitor vertical jump performance in primary school children. Moreover, the data presented in Bland–Altman plots (Figure 2, Figure 3 and Figure 4), show that most of the values are close to the mean of the differences between instruments, thereby representing a high level of agreement [37]. The vertical jump test, as a simple method to quantify the lower limb muscle strength [38], is a part of test batteries used to measure physical ability [39]. The most common tests that are used in studies and by practitioners are SJ and CMJ tests. However, there are different methods and devices for measuring jump height, and the majority of them are expensive and limited to a clinical laboratory.

Our test–retest design in the group of primary school children revealed CMJ and CMJAM height appeared as a reliable assessment outcome (CV < 5% and ICC > 0.90), with wider between-day variability in the children’s SJ outcomes over the two sessions. The current results showed small mean differences of 0.05, 0.04, and 0.02 cm in SJ, CMJ, and CMJAM height for boys and girls together. This is in line with a 4-week repeat analysis of a subsample of CMJ trials that reported a mean difference of 0.04 cm in trained junior athletes [40] and significantly lower compared to difference reported by Stanton et al. [41] (0.43 cm). However, in the study of Stanton et al. [41], the reliability study was conducted on the My Jump app with recreationally active athletes. The concurrent validity of the SJ, CMJ, and CMJAM was assessed by comparing outcome measures to the OptoJump, which is validated for estimating vertical jump. Very large correlations were observed between My Jump 2 app and OptoJump in both the boys (r = 0.98–0.99) and girls (r = 0.97), which supported the validity of the My Jump 2 for assessing a vertical jump in primary school children.

Studies have compared the My Jump and My Jump 2 app with force platform and contact platform measurements on a number of different jumps [17,22,40]. The abovementioned studies showed nearly perfect correlation (r = 0.97–0.99) for CMJ and SJ in trained athletes [22,40], but also for drop jumps (r = 0.94–0.97) in sport science students [17]. Gallardo-Fuentes et al. [22] found a mean difference between devices (0.20 cm) when testing CMJ in both male and female athletes. In one recent study on junior athletes [40], the mean difference in CMJ between devices was 0.59 cm, which is slightly higher than the mean difference found in our study for CMJ (0.20–0.30 cm). Balsalobre-Fernández et al. [20] found higher mean underestimation (1.1–1.3 cm). However, this was probably due to the lower sampling rate (iPhone 5s; 120 Hz), which was improved with the My Jump 2 app. Studies that compared portable measurement devices with force plates showed mean differences for CMJ performance between −1.06 and 11.7 cm [15,42,43]. These differences between the two tools across the literature could be due to the sampling rate and the level of performance [40]. Nevertheless, our study supports the abovementioned results showing jump height estimates are within 1 cm when captured via My Jump 2 on devices capable of video capture at 240 Hz.

Both, the SJ and CMJ are commonly used in laboratory settings [44], but in physical education settings, Sargent jump and the reach test was shown as a predominant test, because of its simplicity and practicality [45,46]. However, the jump and reach vertical jump test is not only focused on the lower limbs but includes both the lower and upper limbs [44,47,48]. Therefore, in order to gain information about leg power performance and reactive strength of the lower limbs [49], SJ and CMJ tests are highly recommended in primary schools, especially concerning the fact that strength is part of health-related fitness, and lower limb strength is rarely assessed. Furthermore, it was expected that the CMJAM would be a confounding variable due to coordinative issues during this jump that can have an impact on jump performances (8−11%) [50,51], as a result of the shoulder, elbow, hip, and ankle muscles working together. However, on the contrary, our results showed surprisingly good results regarding the validity, reliability, and usefulness of CMJAM in primary school children. We could speculate that this was due to the use of the Sargent test and standing long jump that are predominantly used in physical education settings.

Although the validity and reliability of SJ and CMJs have been reported in the literature [44], a great majority of studies were on adult populations and professional athletes with very few reports in youths. Moreover, the instruments that are considered as the gold standard in measuring the vertical jumps are relatively expensive. Therefore, My Jump 2, with its simplicity and practicality, showed as a good low-cost option for measuring s vertical jump compared to reliable and validated Optojump photoelectric cells system. The strength of the study is the use of healthy young students, iPhone X with a 240 Hz high-speed camera, the relatively large number of participants, and field-testing conditions rather than a controlled laboratory space.

A possible limitation of our study was that some participants might not have been familiar with the SJ jump style. Relatively smaller usefulness obtained in SJ may be due to a lack of proper technique among children, whilst previous research was conducted on elite athletes [20,40] with greater experience performing these jumps. In SJ, a participant must not perform countermovement before take-off. This is not a natural jumping technique, so that is why some participants perhaps did a little downward movement before the upward movement. The possible use of a force plate instead of OptoJump would support this argument. Additionally, the evaluator’s inexperience may have conditioned the results.

Additionally, another limitation was that we did not check for the inter-rater reliability because some factors could contribute to differences in scores (experience of the tester, the different variability of scores, testers’ seat positions, and view angle) [52]. Therefore, a future study should include a greater number of observers to compare results measured and to account for possible human error, which could have occurred. Furthermore, the children in our study had an age range from 10 to 14 years, which could be considered when making conclusions as values would likely differ in children of different age ranges, with a larger sample size and with different training or sports backgrounds. Nevertheless, our results support the usage of smartphone apps in measuring a vertical jump in primary school children. Due to its popularity, affordability, portability, and advanced technology, smartphone apps will soon be commonplace for measuring variables associated with physical fitness and health with great accuracy [53].

## 5. Conclusions

The results of this study suggest that My Jump 2 is a valid, reliable, and useful tool for measuring jump height in primary school children. These results have great value having in mind that using valid, reliable, and feasible measurement in the identification of children who are not developing healthy fitness habits is essential. Therefore, My Jump 2 is a low-cost, simple, practical measurement tool that could be used by practitioners and teachers to evaluate the changes in physical fitness with a robust and simple test as SJ, CMJ, and CMJAM.

## Figures and Tables

**Figure 1 ijerph-17-03708-f001:**
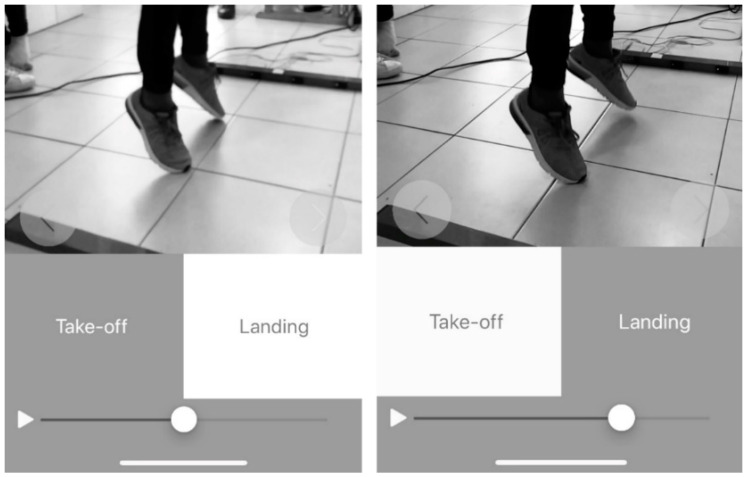
Take-off and landing phase frames on the My Jump 2 app.

**Figure 2 ijerph-17-03708-f002:**
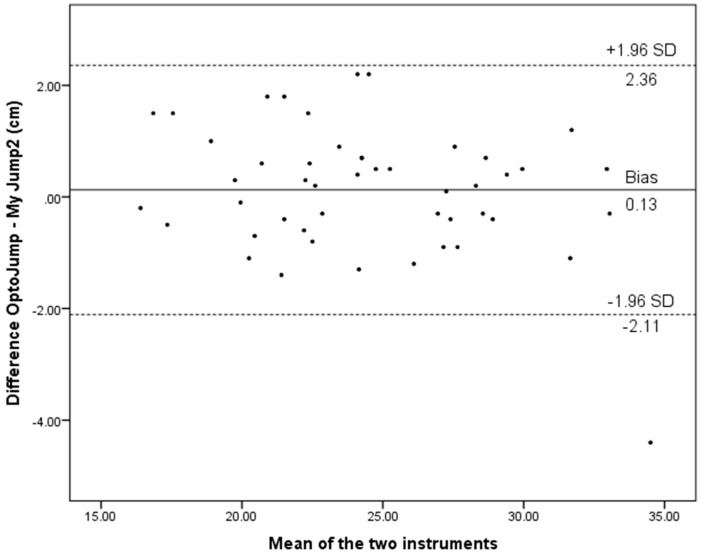
Level of agreement (Bland–Altman) with 95% limits of agreement (dashed lines) and the mean difference (solid line) between My Jump 2 and the OptoJump for SJ in the whole sample.

**Figure 3 ijerph-17-03708-f003:**
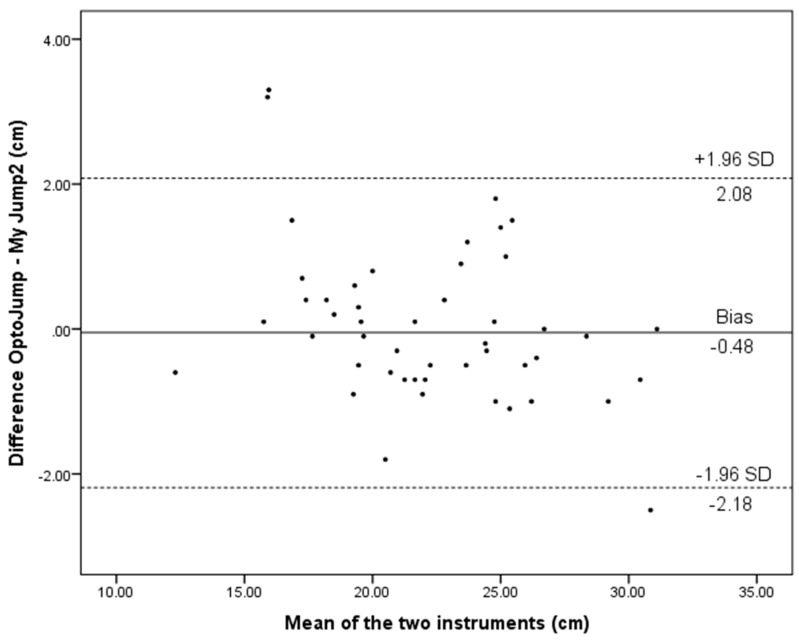
Level of agreement (Bland–Altman) with 95% limits of agreement (dashed lines) and the mean difference (solid line) between My Jump 2 and the OptoJump for CMJ in the whole sample.

**Figure 4 ijerph-17-03708-f004:**
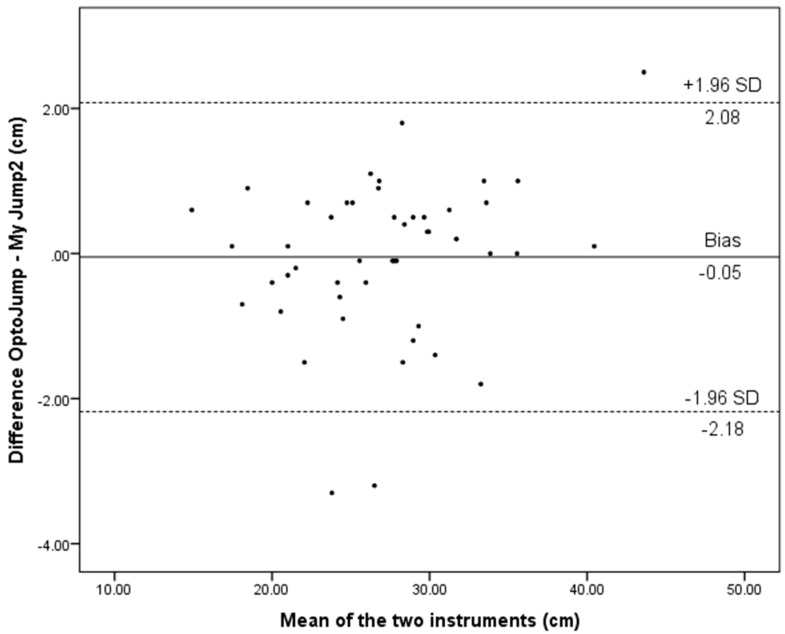
Level of agreement (Bland–Altman) with 95% limits of agreement (dashed lines) and the mean difference (solid line) between My Jump 2 and the OptoJump for CMJAM in the whole sample.

**Table 1 ijerph-17-03708-t001:** Descriptive characteristics.

Variable	Boys (*n* = 26)	Girls (*n* = 22)
Age (years)	12.3 ± 0.8	11.8 ± 0.8
Height (cm)	159.3 ± 13.2	157.3 ± 10.5
Weight (kg)	51.8 ± 18.3	50.6 ± 11.6
Leg length (cm)	96.2 ± 8.5	97.0 ± 6.4

Note: Values are expressed as mean ± SD.

**Table 2 ijerph-17-03708-t002:** Reliability measure for My Jump 2 in primary school children.

	ES	Diff (95%CI)	ICC (95%CI)	TE (95%CI)	CV% (95%CI)	SWC%	Rating
**SJ**							
**All**	−0.12 trivial	0.5 (−0.1; 1.1)	0.88 (0.81; 0.93)	1.5 (1.2; 1.8)	7.6 (6.4; 9.6)	0.8 (3.3%)	marginal
**Boys**	−0.06 trivial	0.2 (−0.8; 1.3)	0.89 (0.75; 0.94)	1.6 (1.2; 2.3)	8.3 (6.3; 12.6)	0.9 (3.7%)	marginal
**Girls**	−0.23 small	0.7 (0.0; 1.4)	0.95 (0.85; 0.97)	0.8 (0.6; 1.4)	4.8 (3.5; 8.1)	0.7 (3.3%)	marginal
**CMJ**							
**All**	−0.10 trivial	0.4 (0.0; 0.8)	0.96 (0.93; 0.97)	1.0 (0.8; 1.3)	4.6 (3.9; 5.8)	1.0 (3.4%)	ok
**Boys**	−0.10 trivial	0.5 (−0.2; 1.2)	0.97 (0.93; 0.99)	1.0 (0.8; 1.5)	4.9 (3.7; 7.3)	1.1 (4.0%)	good
**Girls**	−0.12 trivial	0.2 (−0.6; 1.1)	0.92 (0.77; 0.96)	1.0 (0.8; 1.7)	4.7 (3.4; 7.9)	0.7 (3.1%)	marginal
**CMJAM**							
**All**	−0.10 trivial	0.2 (−0.2; 0.7)	0.97 (0.94; 0.98)	1.1 (0.9; 1.4)	4.9 (4.1; 6.2)	1.2 (3.7%)	good
**Boys**	−0.14 trivial	0.3 (−0.3; 1.0)	0.98 (0.96; 0.99)	0.9 (0.7; 1.4)	4.1 (3.1; 6.1)	1.4 (4.9%)	good
**Girls**	−0.03 trivial	0.3 (−0.5; 1.1)	0.93 (0.81; 0.97)	1.0 (0.7; 1.6)	3.9 (2.8; 6.5)	0.7 (2.2%)	marginal

Abbreviations: SJ, squat jump; CMJ, countermovement jump; CMJAM, countermovement jump free arms; ES, Cohen *d* effect size; Diff, difference; ICC, intraclass correlation coefficient; TE, typical error; CV, coefficient of variation; SWC, smallest worthwhile change.

**Table 3 ijerph-17-03708-t003:** Descriptive statistics and validity analysis based on Pearson’s r.

	My Jump 2	OptoJump	Diff. (95% CI)	ES	r (95% CI)	Rating
**SJ**						
**All**	22.3 ± 4.1	22.2 ± 4.5	0.1 (−1.36; 1.56)	0.02	0.97 ** (0.95; 0.98)	Very large
**Boys**	23.2 ± 4.6	23.1 ± 5.2	0.1 (−2.18; 2.38)	0.02	0.98 ** (0.93; 0.99)	Very large
**Girls**	21.1 ± 3.2	21.3 ± 3.4	−0.2 (−1.87; 1.47)	−0.06	0.97 ** (0.91; 0.99)	Very large
**CMJ**						
**All**	24.5 ± 4.7	24.6 ± 4.3	−0.2 (−1.63; 1.43)	−0.02	0.97 ** (0.95; 0.98)	Very large
**Boys**	25.1 ± 5.5	25.2 ± 4.9	−0.2 (−2.52; 2.32)	−0.02	0.98 ** (0.94; 0.99)	Very large
**Girls**	23.7 ± 3.5	24.0 ± 3.6	−0.3 (−2.10; 1.50)	−0.08	0.97 ** (0.89; 0.99)	Very large
**CMJAM**						
**All**	27.0 ± 5.8	27.2 ± 5.8	−0.2 (−2.17; 1.77)	−0.03	0.99 ** (0.98; 0.99)	Very large
**Boys**	28.2 ± 6.9	28.1 ± 6.8	0.1 (−3.08; 3.28)	0.01	0.99 ** (0.99; 1.0)	Very large
**Girls**	25.8 ± 3.9	26.1 ± 4.0	−0.3 (−2.53; 1.93)	−0.08	0.97 ** (0.91; 0.99)	Very large

Abbreviations: SJ, squat jump; CMJ, countermovement jump; CMJAM, countermovement jump free arms; ES, Cohen d effect size; Diff, difference; r, correlation coefficient; ** *p* < 0.01.

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
