# Peer review of "Validity, Reliability, and Usefulness of My Jump 2 App for Measuring Vertical Jump in Primary School Children"

_ijerph, 2020, doi:10.3390/ijerph17103708_

Round 1
Reviewer 1 Report
The paper describes the quality of the authors' app.
No much scientific novelty. It is a methodology/self-advertisement paper.
The description is rigorous. Figures good have a better format to avoid much space left blank
In my opinion good enough to be published in IJERPH MDPI.
Author Response
Reviewer 1
The paper describes the quality of the authors' app.
No much scientific novelty. It is a methodology/self-advertisement paper.
The description is rigorous. Figures good have a better format to avoid much space left blank
In my opinion good enough to be published in IJERPH MDPI.
Our response: We are thankful for your comments and grateful that you agree with the publication of our paper. We are not the developers of the app. This paper aimed to check the validity of a different target group (primary school children) of an already used app. The app has a practical value for evaluating jump height in primary school children.
We put bigger figures because of better clarity since some numbers on the figures are really small.
Reviewer 2 Report
General comments
The study “Validity, reliability, and usefulness of My Jump 2 app for measuring vertical jump in primary school children” aimed to analyze the validity and the reliability of an application, available for mobile phones, for measuring primary school children jump performance, which is used for predicting their physical fitness. In general, it is very interesting and well-written study. I am in favor of its publication, but some minor issues are presented below.
Specific comments
Abstract:
Please, include the statistical procedures.
Introduction
The introduction is really well written and provides almost all relevant information to comprehend the research problem and the aim of the study. I congratulate the authors for being able to develop a very concise but complete and up-to-date introduction.
My only suggestion is regarding the lines 54-59. You presented important data related to the validity of the MyJump 2 application to different groups. But why a device that has shown a good validity for many different groups would not present for primary school children? Why is this public specific to the point of requiring new validation studies? You should clearly point out the specificities of this public that demand caution when interpreting validation information from other publics.
Materials and Methods
How was the sample estimated? Is there a statistical sample estimation procedure?
Line 106: seems important to point out which version of the iOS was installed in the mobile. Also, which version of the iPhone X was used. Finally, since there are constant updates in both the operating system and the app, seems important to point out when the data collection was made.
How familiarization with the procedures was done? Is there an objective measure of familiarization for each subject? Seems not enough just assuming they were familiar without presenting any data supporting this assumption. In a validation study is mandatory to reduce to the minimum the interference of confounding variables, and the acute learning of the technique is very plausible to this group. Therefore, more details about the familiarization are demanded.
What was the criterion for selecting only 3 jumps? This seems a little arbitrary and the n seems very low for validation studies.
Why was the data analyzed separately to boys and girls? Differences between them are expected? If you are testing a product validity and reliability, I am not sure this split is necessary.
Results
The results are ok. I am just not sure if splitting the data between boys and girls is needed. If not, I think the tables could be reduced to the whole sample, which would make them smaller and easier to read and comprehend.
Discussion
There are no issues.
Author Response
Reviewer 2
General comments
The study “Validity, reliability, and usefulness of My Jump 2 app for measuring vertical jump in primary school children” aimed to analyze the validity and the reliability of an application, available for mobile phones, for measuring primary school children jump performance, which is used for predicting their physical fitness. In general, it is very interesting and well-written study. I am in favor of its publication, but some minor issues are presented below.
Our response: Thank you for your detailed review of our manuscript and for providing some insightful and thought-provoking suggestions to strengthen our manuscript. We feel we have sufficient responses to each of your major concerns listed above, which are further detailed below, and hope that they alleviate the concerns you have regarding the approaches adopted in our manuscript.
Specific comments
Abstract:
Please, include the statistical procedures.
Our response: We addedd the statistical procedure in Abstract.
Introduction
The introduction is really well written and provides almost all relevant information to comprehend the research problem and the aim of the study. I congratulate the authors for being able to develop a very concise but complete and up-to-date introduction.
My only suggestion is regarding the lines 54-59. You presented important data related to the validity of the MyJump 2 application to different groups. But why a device that has shown a good validity for many different groups would not present for primary school children? Why is this public specific to the point of requiring new validation studies? You should clearly point out the specificities of this public that demand caution when interpreting validation information from other publics.
Our response: Thanks for this suggestion, we updated the sentence. Please see lines 53-54.
Materials and Methods
How was the sample estimated? Is there a statistical sample estimation procedure?
Our response:
Shapiro-Wilk test was used to check the data normality, teherfore we used a statistic parametric approach. See Statitical Analysis paragraph.
Line 106: seems important to point out which version of the iOS was installed in the mobile. Also, which version of the iPhone X was used. Finally, since there are constant updates in both the operating system and the app, seems important to point out when the data collection was made.
Our response: Thank you for your suggestions. We have aded the requested informations. Please see Participants section, line 69 and Procedure section lines 85.
How familiarization with the procedures was done? Is there an objective measure of familiarization for each subject? Seems not enough just assuming they were familiar without presenting any data supporting this assumption. In a validation study is mandatory to reduce to the minimum the interference of confounding variables, and the acute learning of the technique is very plausible to this group. Therefore, more details about the familiarization are demanded.
Our response: We are sorry for this omission. We have added more details in the Procedures section, line 77-78.
What was the criterion for selecting only 3 jumps? This seems a little arbitrary and the n seems very low for validation studies.
Our response: We used 3 jump to avoid any fatigue effect, as standard procedure 3 jump are able to perform the best jump performance (https://www.ncbi.nlm.nih.gov/pubmed/26985131)
Why was the data analyzed separately to boys and girls? Differences between them are expected? If you are testing a product validity and reliability, I am not sure this split is necessary.
Results
The results are ok. I am just not sure if splitting the data between boys and girls is needed. If not, I think the tables could be reduced to the whole sample, which would make them smaller and easier to read and comprehend.
Our response
We analyzed separately for 3 reasons:
- to demonstrate the gender effects (https://www.ncbi.nlm.nih.gov/pubmed/26985131)
- to better replicate this study in the future
- available data for a sistematic review or meta-analysis on jump peformance in the future
Discussion
There are no issues.
Our response: Thank you for all your comments.
Reviewer 3 Report
The aim of this research is original, and the methodological approach is valid. Nevertheless, for a study like this one, a larger sample (n=48 is far from being enough) and more homogeneus (there is a mixture of prepubertal and pubertal children) is strongly needed.
Author Response
Reviewer 3
The aim of this research is original, and the methodological approach is valid. Nevertheless, for a study like this one, a larger sample (n=48 is far from being enough) and more homogeneus (there is a mixture of prepubertal and pubertal children) is strongly needed.
Our response: Thank you for your comment and suggestion. To avoid any confusion reader, the goal of this study was to compare the jump performance between two devices, while a double comparison intra-group will be misleading.
Reviewer 4 Report
Attached is the document with some specific suggestions / changes, in order to improve the quality of the paper.

Author Response
Reviewer 4
Very nice paper about as examine the concurrent validity and test-retest reliability of My Jump 2 app compared to a validated OptoJump instrument for measuring jump performance during the squat jump (SJ), countermovement jump (CMJ), and CMJ Free Arms (CMJAM) in primary school children. There are some specific changes and suggestions that should be made to improve the quality of the paper.
Our response: Thank you for your detailed review of our manuscript and for providing some insightful and thought-provoking suggestions to strengthen our manuscript. We feel we have sufficient responses to each of your major concerns listed above, which are further detailed below, and hope that they alleviate the concerns you have regarding the approaches adopted in our manuscript.
General questions
- Insert in the introduction section what were the study hypotheses
Our response: The hypotheses were added. Please see line 63-64.
- Pg02Ln77 - Enter Tanita country and city
Our response: We have added the requested information. The country is Illinois, and the city is Arlington Heights. See line 81.
- Pg03Ln109 - The results may have been conditioned by the evaluator's inexperience.
Our response: We added this information in the Discussion section, where we were talking about the limitations. Please see line 272-273.
- In the statistical analysis section, enter the level of significance established to calculate the results.
Our response: We added the sentence: »The statistical significance was assessed at the p < 0.05 level.« at the end od the Stat. analysis section.
- Put in table 1 the values found with the same number of decimal places
Our response: Thank you for noticing. We have corrected the decimal numbers. See Table 1.
- Insert in the legend of table 3 the level of significance corresponding to the symbol ** for the correlations
Our response: We explained the symbol in the legend of Table 3. Thank you for your remark.
- Pg07Ln191 - Is the figure in the figure correct? Is it not figure 3?
- Pg08Ln199 - Is the figure in the figure correct? Is it not figure 4?
Our response: Thank you for noticing this mistake. We have corrected the text in to Figure 3 and Figure 4. See line
- Pg08Ln208 - This paragraph should be at the beginning of the discussion, since the first paragraph of the discussion is to remember the objective of the study
Our response: Thank you for your suggestion. We have moved the paragraph. See Discussion section line 207-214.
Round 2
Reviewer 3 Report
I have not changed my mind regarding my first comments. Small sample size and heterogeneous even for the tricky change made by the authors.